# Superior High Transistor’s Effective Mobility of 325 cm^2^/V-s by 5 nm Quasi-Two-Dimensional SnON nFET

**DOI:** 10.3390/nano13121892

**Published:** 2023-06-20

**Authors:** Pheiroijam Pooja, Chun Che Chien, Albert Chin

**Affiliations:** Department of Electronics Engineering, National Yang Ming Chiao Tung University, Hsinchu 300, Taiwan

**Keywords:** high mobility, thin-film transistors, SnON, SnO_2_, density functional theory

## Abstract

This work reports the first nanocrystalline SnON (7.6% nitrogen content) nanosheet n-type Field-Effect Transistor (nFET) with the transistor’s effective mobility (µ_eff_) as high as 357 and 325 cm^2^/V-s at electron density (Q_e_) of 5 × 10^12^ cm^−2^ and an ultra-thin body thickness (T_body_) of 7 nm and 5 nm, respectively. At the same T_body_ and Q_e_, these µ_eff_ values are significantly higher than those of single-crystalline Si, InGaAs, thin-body Si-on-Insulator (SOI), two-dimensional (2D) MoS_2_ and WS_2_. The new discovery of a slower µ_eff_ decay rate at high Q_e_ than that of the SiO_2_/bulk-Si universal curve was found, owing to a one order of magnitude lower effective field (E_eff_) by more than 10 times higher dielectric constant (κ) in the channel material, which keeps the electron wave-function away from the gate-oxide/semiconductor interface and lowers the gate-oxide surface scattering. In addition, the high µ_eff_ is also due to the overlapped large radius s-orbitals, low 0.29 m_o_ effective mass (m_e_*) and low polar optical phonon scattering. SnON nFETs with record-breaking µ_eff_ and quasi-2D thickness enable a potential monolithic three-dimensional (3D) integrated circuit (IC) and embedded memory for 3D biological brain-mimicking structures.

## 1. Introduction

Modern processors, with over 100 billion transistors, are among the most complex systems. To meet the ever-changing demand for small and high-performance devices, processor transistor density and performance must be increased. Therefore, Moore’s law must be preserved, i.e., the Metal-Oxide-Semiconductor Field-Effect Transistor (MOSFET) must continue shrinking in size. The conventional MOSFET is a surface channel device. In long-channel conventional MOSFETs technology, the characteristics of the transistor were at par with the essential speed as well as power requirements. In this era of electronics, power saving and low leakage are more crucial compared to an increase in speed. The drive current rises with the new generation of transistor. However, there is also a tremendous enhancement in the subthreshold leakage current, which results in an increase in the power consumption [1]. Moreover, in FETs with a small channel length, the depletion regions underneath the source and drain cause degraded FET’s off-state leakage (I_OFF_), poor sub-threshold slope (SS), and threshold voltage (V_T_) reduction by drain-induced barrier lowering. To overcome those short-channel effects, a thin body thickness (T_body_) Si-on-Insulator (SOI) was invented. The SOI can use a substrate bias to improve the gate electrostatic control on channel carriers. When transistor sizes grew smaller in conventional planar MOSFETs with a 1.2 nm SiO_2_ gate oxide, the market required a significant innovation to retain performance while limiting short channel effects and power in advanced technologies, as the DC leakage in SiO_2_ is intolerably high. It was necessary to create a gate dielectric that could be substituted with SiO_2_, one that was thick enough to block direct electron tunneling through it but permeable enough to allow the electric field of the gate to enter the channel. Therefore, the solution was to use high dielectric–constant (high-κ) dielectric, which is a dielectric material that has a higher dielectric permittivity than SiO_2_. Although the use of high-κ dielectrics with metal gates increased the lifetime of planar MOSFET by decades, it became necessary to introduce new devices beyond the 28 nm technology node to address the problems with traditional MOSFET [2]. Tremendous efforts have been made to reduce oxide thickness (t_ox_) and increase ε_ox_ to further decrease the gate length while retaining sufficient gate controllability. Yet, the Si T_body_ of SOI requires continuously thinning down to improve the short channel effects, which cause a technology challenge. One simple solution is to form the three-dimensional FinFET that has an even thinner Si Fin down to 6 nm T_body_ thickness. Both the sidewalls and top surface of Fin are covered by gate oxide and metal gate, which have better gate electrostatic control of the channel carriers than SOI. Therefore, the FinFET has been applied to 22 to 3 nm technology nodes rather than using SOI. Figure 1 shows the FET’s technology flow. The figure shows the bulk MOSFET, SOI MOSFET, and three-dimensional FinFET. 

The continuous downscaling decreases the transistor’s source and drain distance and causes lowered drain voltage (V_D_) and power consumption of V_D_I_D_/2, where I_D_ is the drain current. The ultimate V_D_ downscaling is limited by the voltage drop in the sub-threshold region, which has an idea SS of 60 mV/dec. Although the SS can be improved by using the charges in ferroelectric gate dielectric [3], the relatively large thickness and crystallized high-κ gate dielectric are the major concerns to integrate into highly scaled FinFET and nanosheet FET. On the other hand, a high V_D_ is required to deliver enough output power for wireless communication [4]. The highly scaled FinFET and nanosheet FET cannot sustain the high V_D_ that will cause the device to break down. Fortunately, the Vacuum Nano-Triode device in the Nothing-On-Insulator (NOI) configuration may overcome this challenge by operating at a relatively high V_D_ [5,6]. This transistor showed excellent performance up to 4 THz, which is crucial for sixth-generation (6G) wireless communication. For logic application, further research and development to lower the V_D_ and V_G_ to less than 1 V is required for an NOI transistor. 

Nanosheet transistors are the best solution to overcome these challenges of FinFET scaling, enabling higher drive currents [7,8]. The nanosheet FETs are suitable for high computing needs due to their compatibility with various single-crystal materials such as Si, SiGe, two-dimensional (2D) MoS_2_ and WS_2_, among others. The downscaling of Si nanosheet complementary FET is planned to 1 nm node, but further shrinking of the device is limited by the implementation of 2D materials and hyper-numerical-aperture (NA) extreme-ultraviolet (EUV) lithography. Unfortunately, there is no known solution to form a defect-free and uniform monolayer 2D material over the 12-inch wafer. The rapidly increasing cost and huge power consumption are the major bottlenecks to realizing a hyper-EUV lithography system. Those downscaling barriers may be overcome by the monolithic three-dimensional (3D) structure [9,10,11] that mimics the bio-brain. In addition, monolithic 3D integrated circuits (ICs) can provide better performance of higher operating frequencies and lower power consumption than their 2D counterparts [10]. Yet the poor µ_eff_ for a transistor made on the backend dielectric of an IC is the basic challenge. Previously, we reported on the high field-effect mobility (µ_FE_) of SnO_2_ [9,12,13] and SnON FET [14], but the µ_eff_ is the required important data for transistors. The µ_eff_ can give crucial information on electron-scattering mechanisms over the wide range of inversion charge (Q_e_). The Q_e_ or gate voltage (V_G_)-dependent µ_eff_ is also essential for device modeling used for IC design. In this report, we measure the transistor output current over a wide range of V_G_, equivalent to a Q_e_ close to 1 × 10^13^ cm^−2^, to analyze the device-scaling mechanism. The major findings beyond our previous published paper [14] are the much lower µ_eff_ decay rate at high E_eff_ than SiO_2_/Si, high-κ/InGaAs, high-κ/2D MoS_2_ nFETs, etc. This is the new discovery that was never reported in any FET device. In order to deliver a high transistor output current for an FET and drive the IC speed quickly, preserving the high µ_eff_ at a high Q_e_ is critical. The physical limitation of a MOSFET is that the µ_eff_ degrades monotonically with increasing charge density. However, the MOSFET must be biased at high charge density to deliver a high output current. For the first time, this fundamental restriction is overcome by using a higher κ and high µ_eff_ channel. The nanocrystalline SnON n-type FET (nFET) has a µ_eff_ value as high as 325 cm^2^/V-s at 5 × 10^12^ cm^−2^ electron density (Q_e_) and 5 nm nanosheet body thickness (T_body_). At the same T_body_, this µ_eff_ is significantly higher than single-crystalline Si, InGaAs, 2D MoS_2_, 2D WS_2_ and 2D WSe_2_. The high µ_eff_ is due to the >10× higher κ value of SnO_2_ than other semiconductor materials of Si, GaAs, InP, GaN and SiC, which can lower the channel effective field (E_eff_) by >10× even at high Q_e_. In addition, the small 0.29 m_o_ effective mass (m_e_*), large overlapped s-orbitals and low phonon scattering may also play important roles to increase the mobility, although the µ_eff_ depends on both extrinsic and intrinsic scattering mechanisms and will be discussed in the following sessions. The N^3−^ anions having a higher p orbital energy can move up the valance band (E_V_) from first-principle QM calculation, and the oxygen vacancy levels (V_o_) residing in the channel layer are reduced to improve the µ_eff_. The 3D 400 °C process of SnON does not require a single crystal substrate; thus, the energy consumption is many orders of magnitude lower than today’s single-crystal Si wafer. The record-high µ_eff_ and quasi-2D thickness SnON nFET suggest potential monolithic three-dimensional (3D) and embedded dynamic random access memory (DRAM) to mimic the 3D bio-brain structure. 

## 2. Materials and Methods

The bottom-metal-gate/high-κ/[SnON or SnO_2_] nFETs were made by depositing a 50 nm TaN as the bottom gate using reactive sputtering. Then, a 45 nm high-κ HfO_2_ and 3 nm SiO_2_ were deposited as a gate dielectric using an electron-beam evaporator and annealed at 400 °C in an oxygen environment for 30 min using a furnace. Furthermore, a SnON or SnO_2_ channel layer were deposited by reactive sputtering using a Sn target (purity 99.99%) followed by post-annealing at 400 °C. The Sn sputter power, argon flow rate and process pressure are fixed at 30 W, 24 sccm and 7.6 × 10^−3^ torr, respectively. The O_2_ flow rate is fixed at 20 sccm for the SnO_2_ channel layer, while 7.6% nitrogen content (30 sccm of Nitrogen) was used for the deposition of the SnON channel layer. The source-drain electrodes of 80 nm thick Al were deposited and patterned using a thermal coater. The fabricated nFET has a channel length of 50 μm and width of 500 μm. The material properties of SnON and SnO_2_ were studied using first-principle QM calculations [15]. The Broyden–Fletcher–Goldfarb–Shanno (BFGS) minimization technique has been used to optimize the crystal structure [16]. It was performed using the self-consistent field approach, which has a convergence precision of 1 × 10^−8^ eV/atom. This study made use of the generalized gradient approximation (GGA) with local density approximation plus the U (LDA + U) approach. The energy cutoff for enlarging the plane wave basis set was set at 430 eV, and the Brillouin zone was sampled using the Monkhorst–Pack k-point approach with the k-points (6 × 6 × 5) [17]. The electrical characterization of the nFET device was analyzed using the HP4155B semiconductor parameter analyzer with the help of a probe station. 

## 3. Results

Figure 2 displays the cross-sectional transmission electron microscopy (TEM) image of the 5 nm SnON/SiO_2_/HfO_2_ stack on a Si substrate. A nanocrystalline uniform SnON layer of 5 nm ultra-thin thickness was observed. To enlarge the I_ON_, a gate insulator with high-κ [18] HfO_2_ was employed to reduce the operating voltage. Between the channel and gate dielectric, SiO_2_ with 3 nm thickness was deposited to limit the remote phonon scattering occurring from the high-κ gate dielectric [19].

Using first-principle calculations based on density functional theory, the density of state (DOS) for SnO_2_ and SnON were examined as shown in Figure 3a,b, respectively. For convenience of analysis, the valence band maximum (VBM) was adjusted to zero. The lower conduction states close to the conduction band minimum (CBM) in SnO_2_ and SnON were primarily produced from Sn 5s orbitals [20], while the localized states immediately above the VBM in SnON had a predominance of N 2p character. The N states in the valence band, principally N 2p character, are the main cause of the bandgap reduction in SnON. SnO_2_ and N_2_-doped SnO_2_ have effective electron masses (m_e_*) of 0.41 m_o_ and 0.29 m_o_, respectively, where m_o_ is the free electron mass which is reported in our previous work [14]. The m_e_* for SnON is evidently smaller than SnO_2_, which could result in a larger µ_eff_. 

Figure 4a–c depict the transistor’s drain current versus drain voltage (I_D_–V_D_) characteristics at various V_G_ for SnO_2_ and SnON nFETs with T_body_ of 5 nm and 7 nm. A clear pinch-off and good current saturation were measured. The SnON nFETs displayed higher I_D_ compared to the control SnO_2_ device. Because the metal gate/high-κ was made at the same run with identical gate oxide capacitance, the only reason to cause a significantly higher I_D_ at the same V_G_–V_T_ of SnON nFET is due to the higher µ_eff_.

Figure 5a,b display gate current versus gate voltage (I_G_–V_G_) and I_D_–V_G_ transfer characteristics at a V_D_ = 0.1 V for SnON nFETs with T_body_ values of 5 and 7 nm, respectively. A large on-current/off-current (I_ON_/I_OFF_) is achieved in 5 nm T_body_ thickness, which is important for IC application. For accurate µ_eff_ extraction, a fat FET (long channel FET) [21] made in IC fabs must be used to lower the difference between physical and electrical gate length, where the source and drain depletion regions can decrease the electrical gate length. This is the reason why mA is used for the *Y*-axis rather than mA/μm. 

The FET’s scattering mechanism is further analyzed by the µ_eff_ as a function of Q_e_. The µ_eff_ values of FET are calculated according to the conventional metal-oxide-semiconductor (MOS) FET model [22,23,24]:(1)$${\;µ}_{\mathrm{eff}}=\frac{L_{G}}{W_{G}\;}\frac{{\mathrm{dI}}_{\mathrm{DS}}}{{\mathrm{dV}}_{\mathrm{DS}}}\;\frac{1}{C_{\mathrm{ox}}\left(V_{\mathrm{GS}}-V_{T}\right)}\;,\;$$
where L_G_ and W_G_ are the length and width of the conducting channel, respectively, and Cox is the gate-oxide capacitance. As shown in Figure 5c, at low to medium Q_e_, the nFET’s µ_eff_ of SnO_2_ is significantly lower than that of the SnON one. The SnO_2_ nFET shows much faster µ_eff_ degradation with increasing Q_e_. Although the oxide charges in a high-κ dielectric are responsible for lower µ_eff_ than the conventional SiO_2_ gate dielectric [25,26,27,28], such a µ_eff_ reduction is most significant at high Q_e_ rather than at low Q_e_. It is reported that the µ_eff_ at low E_eff_ or Q_e_ is due to Coulomb scattering from charged impurities [24]. The potential reason for such a larger µ_eff_ of SnON nFET than that of SnO_2_ may be related to the lower charged V_o_. By injecting non-oxide nitrogen anions, SnON can lower the defect trap densities. This allows for the removal or passivation of Vo through substitutional alloying with N^3−^ to improve the μ_eff_, as seen in Figure 6. Similar observations were also found with ZnON [29]. It is well known that the transition SiO_x_ between Si and SiO_2_ gives a positive fixed oxide charge, which is primarily due to structural V_o_ defects in the oxide layer. Such a positive V_o_ charge close to the valence band in SnON may be lowered by an extra N-band, as shown in the DOS of Figure 3b. 

Figure 7a further plots 1/µ_eff_ versus Q_e_. The 1/µ_eff_ has a linear relationship with Q_e_.
1/µ_eff_ = kQ_e_, (2)
where k is the proportional constant. The inversely linear relationship between µ_eff_ and Q_e_ is exactly the same as the µ_eff_ dependence on ionized impurity concentrations [24]. This confirms that the charged V_o_ in SnO_2_ is the major reason to cause Coulomb scattering. The large slope in the low Q_e_ is related to charged V_o_ scattering in SnO_2_ that is lowered by adding N^3-^ anions. We further compare the µ_eff_–Q_e_ dependence using Equation (2) for universal SiO_2_/bulk-Si, SiO_2_/Si-on-Insulator (SOI), high-κ/SnO_2_, and high-κ/SnON nFETs. As shown in Figure 7b, µ_eff_ values as high as 357 and 325 cm^2^/V-s are achieved at Q_e_ of 5 × 10^12^ cm^−2^ and T_body_ of 7 and 5 nm, respectively. At 1 × 10^13^ cm^−2^ Q_e_, an ultra-thin 5 and 7 nm thickness, the µ_eff_ of high-κ/SnON nFET is 85% and 95% of universal SiO_2_/bulk-Si nFET. The µ_eff_ scattering mechanism of SiO_2_/bulk-Si nFET at low, medium, and high E_eff_ is due to Coulomb, phonon, and surface scattering, respectively. The universal µ_eff_ of SiO_2_/bulk-Si nFET depends on standard Q_e_^−0.3^ in medium Q_e_, which becomes Q_e_^−0.6^ dependence at high Q_e_ to 1 × 10^13^ cm^−2^. However, the µ_eff_ decay rate of high-κ/SnO_2_ and high-κ/SnON nFETs at high Q_e_ is much slower than universal SiO_2_/bulk-Si and thin-body SOI nFETs [30].

To understand such abnormal slow μ_eff_ dependence on Q_e_, we further measured the dielectric constant, κ of 5 nm SnO_2_. Figure 8 shows the measured capacitance under various voltages at 1 kHz. The SnO_2_ has a κ of 123, which is >10× larger than major semiconductors of Si, GaAs, InP, GaN, SiC, etc. [31,32,33,34,35]. This high κ value is also close to the reported data in the literature [36]. The novel discovery μ_eff_ dependence on Q_e_^−0.30^ at a high Q_e_ range is due to the >10× higher κ value to keep a high-κ/SnON nFET at the medium E_eff_ range. Here, the E_eff_ is proportional to Q_e_:(3)$${\;E}_{\mathrm{eff}}=\frac{\left|Q_{\mathrm{semi}}\right|}{{Ɛ}_{\mathrm{semi}}}=\frac{1}{{Ɛ}_{\mathrm{semi}}}\left(\frac{\left|Q_{e}\right|}{n}+\left|{\;N}_{\mathrm{dep}}\right|\right)\approx \frac{1}{{Ɛ}_{\mathrm{semi}}}\left(\frac{\left|Q_{e}\right|}{n}\right)@{\;\mathrm{high}\;Q}_{e},$$

The ε_semi_ equals ε_0_κ, where ε_semi_ and ε_0_ are the permittivity of the semiconductor and free space, respectively. N_dep_ is the depletion charge of charged impurities in doped Si or charged V_o_ in major oxide semiconductors. The *n* factor in SiO_2_/bulk-Si is equal to 2 and 3 for nMOSFET and pMOSFET, respectively. This equation is basically Gauss’s law. The Gauss law is one of Maxwell’s equations [37], which cannot be changed in an ultra-thin T_body_ device. This is exactly the reason why this equation has been widely used for 2D material FETs [38]. The significantly much higher κ value than most of the commercial semiconductors of Si, GaAs, InP, GaN and SiC allows the channel electrons to keep a low E_eff_. This in turn keeps the electron wave-functions in the conduction channel [39] away from the gate-oxide/semiconductor interface and decreases the gate-oxide surface scattering. The carrier transport in ultra-thin body or 2D materials is determined by both the intrinsic mobility of phonon scattering and Coulomb scattering from the charge impurities and defects, the extrinsic effects of remote phonon scattering from high-κ dielectrics, and the surface roughness scattering from the oxide/semiconductor interfaces. For InGaAs nFET at a relatively thick T_body_ larger than 20 nm [40], the μ_eff_ is dominated by intrinsic phonon scattering. Therefore, the μ_eff_ of InGaAs nFET is higher than that of Si due to the smaller m_e_*. However, for a thin InGaAs T_body_ less than 20 nm, the extrinsic scattering of interface defects limits the μ_eff_ [40]. In this report, the µ_eff_ values of a thin T_body_ of 7 and 5 nm are still higher than those of a Si and InGaAs nFET. The reason can only be ascribed to the superb intrinsic property of >10× smaller E_eff_ to lower the interface scattering, smaller m_e_*, and high phonon limited mobility. The device modeling of this record high µ_eff_ nFET may be developed by future researchers, as such figures are typical for the past InGaAs FET [41,42,43] and 2D materials FETs [44,45].

It is important to notice that the μ_eff_ values of SnON nFET are the highest values among all the oxide-based semiconductors. This is due to the smaller m_e_* and larger phonon energy (E_op_) [46], which lead to a high μ_eff_:(4)$${µ}_{\mathrm{op}}\;\alpha \;\frac{1}{{\left(\frac{m_{e}^{*}}{m_{0}}\right)}^{\frac{3}{2}}}\;\frac{\exp\left(\frac{E_{\mathrm{op}}}{\mathrm{kT}}\right)-1}{{\left(\frac{E_{\mathrm{op}}}{\mathrm{kT}}\right)}^{\frac{1}{2}}}$$

The E_op_ is higher than ZnO, GaN, and SiC [47,48,49,50].

The total μ_eff_ can be expressed as:(5)$$\frac{1}{{µ}_{\mathrm{total}}}=\frac{1}{{µ}_{\mathrm{intrinsic}}}+\frac{1}{{µ}_{\mathrm{extensic}}}=\frac{1}{{µ}_{\mathrm{Vo}}}+\frac{1}{{µ}_{\mathrm{op}}}+\frac{1}{{µ}_{\mathrm{high}-k}}+\frac{1}{{µ}_{\mathrm{sr}}}$$

Here, the µ_Vo_ is the FET’s mobility that is limited by the charged V_o_. This µ_Vo_ is extremely important at low to medium Q_e_, as shown in Figure 5c. In ultra-thin body 2D materials, the carrier transport is determined by phonon scattering from the dielectrics and Coulomb scattering from charged defects such as vacancies [51]. Ma and Jena et al. predicted that high-κ dielectrics provide an effective screening of the charge impurities, leading to high Coulomb-limited mobility [52]. Moreover, owing to the low formation energy of the chalcogen vacancy, a large amount of sulfur vacancies is commonly observed in synthesized 2D MoS_2_, which can induce short-range scattering and degrade carrier mobility [53]. Thus, Equation (4) is also derived for 2D systems. In addition, the excellent matching of Equation (4) with measurements is also reported for SnO_2_ nFET with a 5 nm channel thickness [46].

The radius of the s-orbital increases with the increasing principle quantum number n with n^2^ dependence, so the overlapping s-orbitals are stronger for SnO_2_ than for ZnO [20]. The theoretical background of high mobility in a metal-oxide semiconductor is due to the overlapped s-orbitals [54]. The larger s-orbitals and the stronger overlapping of electron clouds lead to high mobility. We have earlier reported that in SnON, the localized states just above the valence band maximum (VBM) have a predominant N 2p character and the lower conduction states near the conduction band minimum (CBM) were mostly derived from Sn 5s orbitals, which results in high electron mobility in SnON [14]. This explains why the mobility of SnON nFET is significantly larger than that of ZnO. 

Table 1 compares the device performances. The wide energy bandgap (E_G_) nanocrystalline SnON nFET has the highest µ_eff_ among single-crystal Si, InGaAs, 2D MoS_2_, and 2D WS_2_. It is noticed that the next 2 nm node commercial nanosheet nFET will use single-crystalline Si with a T_body_ of 7 nm, since the µ_eff_ decreases with decreasing T_body_ with a T_body_^6^ dependence [55]. The µ_eff_ of high-κ/SnON nFETs is 2.7 times higher than that of Si nFET at the same 5 nm T_body_, which could be used for downscaling the nanosheet T_body_. The wide-E_G_ SnON also leads to large I_ON_/I_OFF_, as shown in Figure 5a. 

The searching for high µ_eff_ material nFET leads to extensive research on high-mobility InGaAs nMOSFET [41]. The reason why the material failed to be implemented into manufacture is due to the relatively inferior oxide/semiconductor interface, which caused µ_eff_ degradation in thin T_body_ rather than the enhanced tunneling. For a T_body_ value less than 20 nm, the g_m_ and g_m_/T_body_ of Si FinFET are still better than those of InGaAs FinFET [40]. In the InGaAs FET [40,41,42] and 2D material FET [56] evolution, a long gate length device was first made to investigate the intrinsic property, such as µ_eff_, I_on_/I_off_ and SS. The downscaling of InGaAs nFET took a decade-long study, until the µ_eff_ degraded fast with decreasing ultra-thin T_body_. After the record-high µ_eff_ is reported, researchers and engineers in IC fabs will follow up to study the small gate length devices and the potential to be implanted in the gate all around (GAA) nanosheet FET.

Because the remarkably high µ_eff_ SnON nFET is the new data, there is no modeling on the experimental data reported so far. In the scientific field of semiconductor devices, the experiments are carried out before mobility modeling. The modeling work following experiments can be evident from past high-mobility InGaAs nFET development. The superb µ_eff_ in a 5 nm ultra-thin T_body_ will attract modeling experts in the future works. It is well known that the device modeling is developed after MOSFET fabrication in the IC industry, such as the widely used Berkeley Short-channel IGFET Model (BSIM). In this model, there are many fitting parameters to be measured experimentally in addition to physically based equations. As the devices become smaller in each technology node by Moore’s law, new versions of device models are developed to accurately reflect the transistor’s behavior. Therefore, the BSIM model has changed continuously for the past three decades. Such device modeling requires years of experience from both academic and IC fabs’ team works, which is beyond our group’s capability. Similar device modeling followed by this record-high µ_eff_ nFET may be developed later by theoreticians, as these results are typical for the past InGaAs FET [40,41,42,43] and 2D materials FETs [44,45].

## 4. Conclusions

In this work, we demonstrated record-high µ_eff_ 5 nm T_body_ nFETs, made on IC’s backend for monolithic 3D usage. For the first time, the µ_eff_ of 325 cm^2^/V-s at 5 × 10^12^ cm^−2^ Q_e_ is 2.7 times higher than that of Si nFET at the same T_body_ of 5 nm. This was achieved using a wide-E_G_ 5 nm quasi-2D SnON channel processed at 400 °C. Such a high FET’s µ_eff_ is due to the smaller 0.29 m_o_, overlapped large-radius s-orbitals, and low polar optical phonon scattering. In addition, a smaller µ_eff_ decay rate than SiO_2_/bulk-Si nFET at high Q_e_ was found, owing to the <10× E_eff_ by >10× higher κ value. The record-high µ_eff_ SnON nFETs formed on IC’s backend signal empowering technology for monolithic 3D ICs.

## Figures and Tables

**Figure 1 nanomaterials-13-01892-f001:**
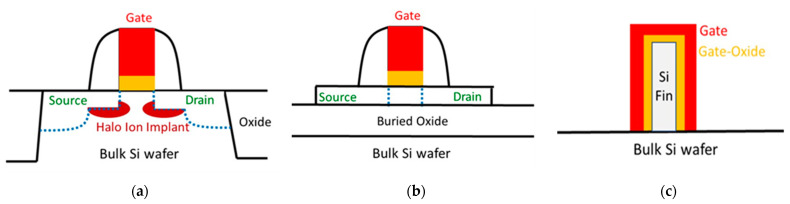
The evolution of device structure from (**a**) planar MOSFET on bulk Si wafer, (**b**) planar MOSFET ultra-thin body SOI, and (**c**) 3D FinFET.

**Figure 2 nanomaterials-13-01892-f002:**
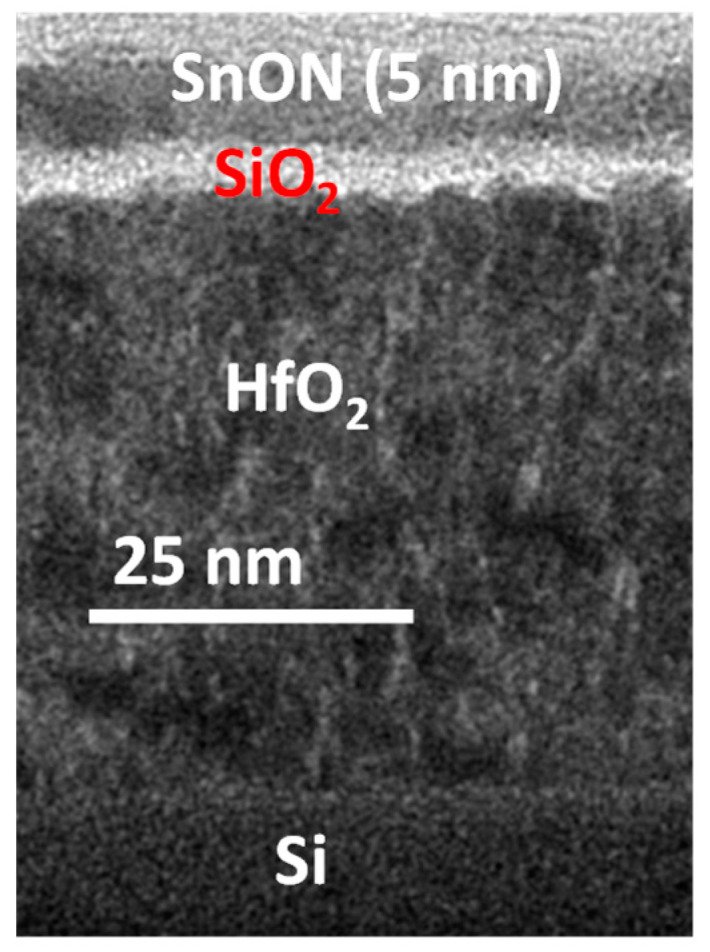
TEM image of the 5 nm-SnON/SiO_2_/HfO_2_ stack.

**Figure 3 nanomaterials-13-01892-f003:**
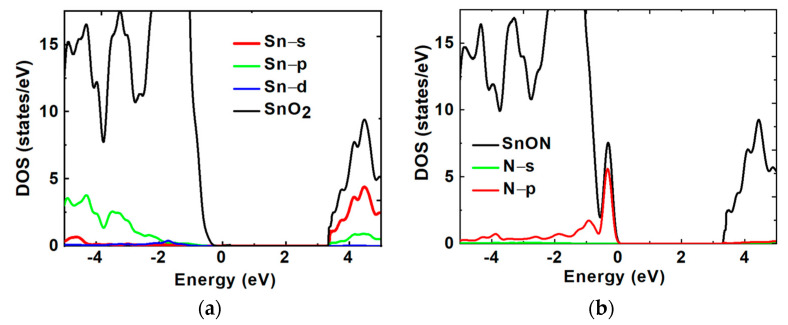
(**a**) DOS of Sn in SnO_2_ and (**b**) DOS of N in SnON calculated using first-principle density functional theory.

**Figure 4 nanomaterials-13-01892-f004:**
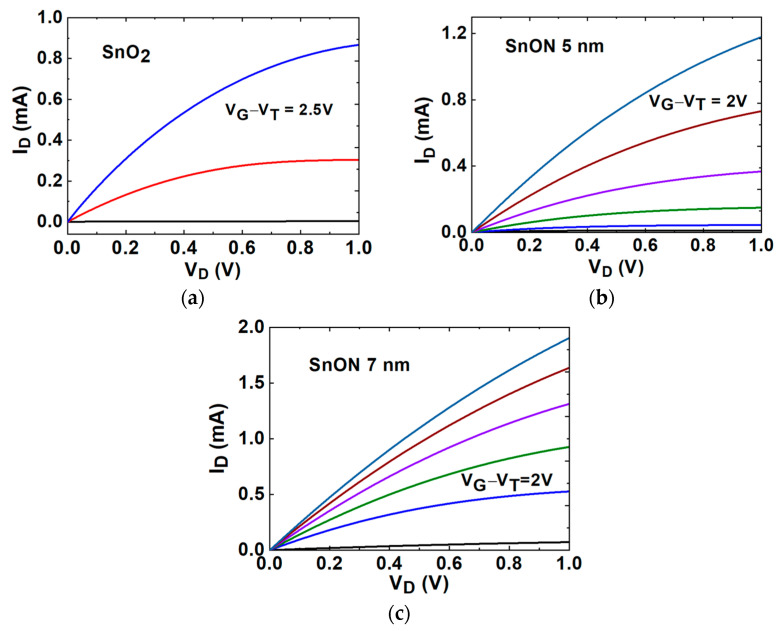
I_D_-V_D_ output characteristics for (**a**) TaN/HfO_2_/5 nm-SnO_2_ nFET, (**b**) TaN/HfO_2_/5 nm-SnON nFET, and (**c**) TaN/HfO_2_/7 nm-SnON nFET.

**Figure 5 nanomaterials-13-01892-f005:**
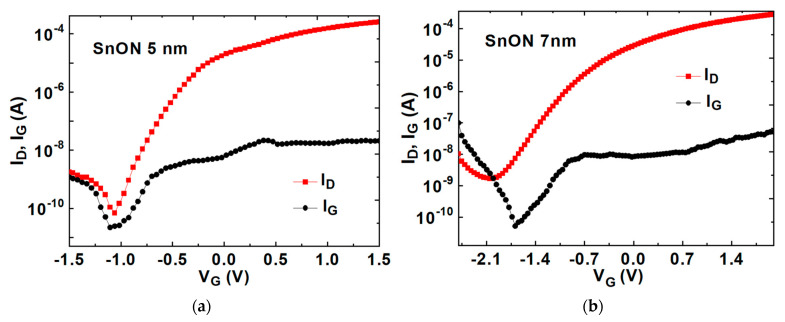
I_G_–V_G_ and I_D_–V_G_ transfer characteristics for (**a**) TaN/HfO_2_/5 nm SnON nFET and (**b**) TaN/HfO_2_/7 nm SnON nFET; and (**c**) μ_eff_ versus *Q_e_* for 5 nm SnO_2_ and SnON nFETs (The dashed lines are used to check the μ_eff_ dependence on Q_e_).

**Figure 6 nanomaterials-13-01892-f006:**
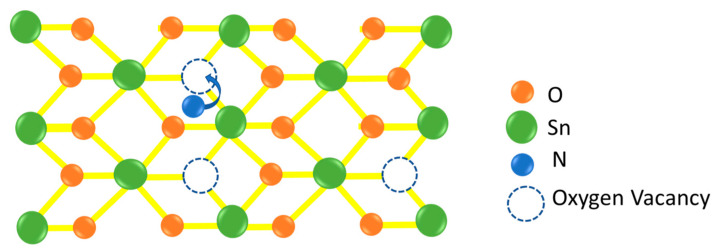
Diagrammatic sketch of substitutional alloying of oxygen vacancy with nitrogen atoms.

**Figure 7 nanomaterials-13-01892-f007:**
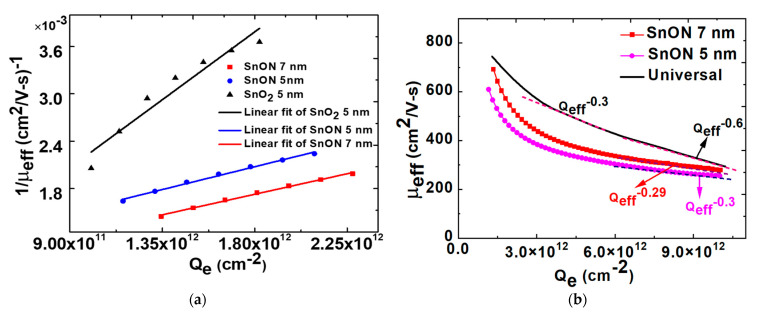
(**a**) 1/μ_eff_ versus Q_e_ plot for 5 nm SnO_2_, 5 nm SnON and 7 nm SnON nTFTs and (**b**) μ_eff_ versus Q_e_ with different channel thickness of SnON nFET and comparison with universal nFETs (The dashed lines are used to fit and check the μ_eff_ dependence on Q_e_).

**Figure 8 nanomaterials-13-01892-f008:**
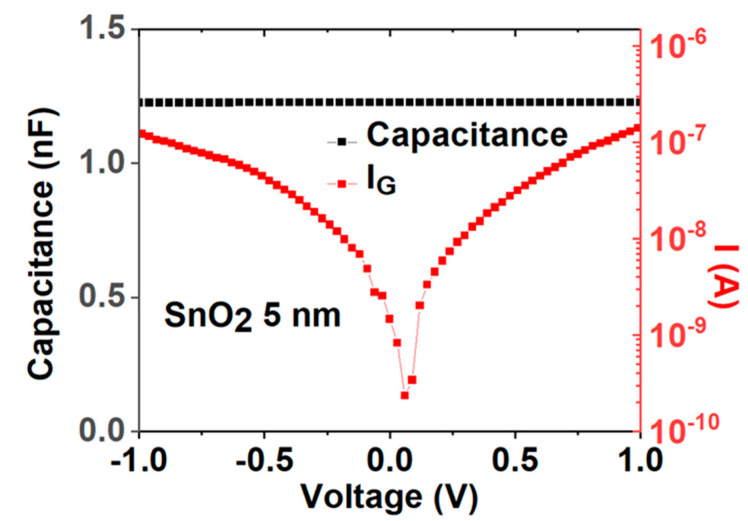
C-V and I-V plot for Ni/SnO_2_/Ni capacitor.

**Table 1 nanomaterials-13-01892-t001:** Comparisons of 2D semiconductor performances with our present work at Q_e_ of 5 × 10^12^ cm^−2^.

Semiconductor Material	E_G_ (eV)	m_eff_ (m_o_)	Dielectric Const. κ	µ_eff_ (cm^2^/V-s) @5 nm
SnON (this work)	~3.3	~0.29	123	325
Si [38]	1.12	1.08	11.7	120
MoS_2_ [38]	1.8	~0.5	4~8 (2~5 layers)	184
WS_2_ [38]	1.4	0.33	-	234
InGaAs [38]	0.75	0.042	12.9	200

## Data Availability

The data presented in this study are available on request from the corresponding author. The data are not publicly available due to privacy.

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
