# Peer review of "Superior High Transistor’s Effective Mobility of 325 cm2/V-s by 5 nm Quasi-Two-Dimensional SnON nFET"

_nanomaterials, 2023, doi:10.3390/nano13121892_

Round 1
Reviewer 1 Report
The paper reports a nanocrystalline SnON (7.6 % Nitrogen content) nanosheet n- 7 type Field-Effect Transistor (nFET) with transistor’s effective mobility (µeff) as high as 357 and 325 8 cm2 /V-s at electron density (Qe) of 5×1012 cm-2 and ultra-thin body thickness.
The paper is well written is valuable, but some informations are missing, and it can be improved by some additional informations, after my opinion.
Authors can provide much more details about the technology including some successive figures of the technological flow. What SOI technique was used for 5nm, 7nm.
How was measured and correlated Qe charge and mobility meff?
It is missing also a SEM/TEM image of the nano-transistor section to prove the thicknesses.
In Introduction, authors claim the FinFET evolution till 1-3nm. I request authors to inset also o brif discussion about another exponent - the NOI transistor citing Ref: (i) Vacuum nano-triode in Nothing-On-Insulator configuration working in Terahertz domain, IEEE Journal of the Electron Devices Society, 6(1), pp. 1115-1123, 2018, DOI: 10.1109/JEDS.2018.2868465
(ii) Manufacturing of a Nothing On Insulator Nano-Structure with two Cr/Au Nanowires Separated by 18 nm Air Gap, Nanotechnology, 31(27), pp.1-9, 2020, https://dx.doi.org/10.1088/1361-6528/ab7c45
After these brief revision I recommend the paper publishing.
Author Response
Dear Reviewer 1,
We have responded to your important comments, point by point, and have included them in the revised manuscript (shown in green color). Your comments and our replies are in the following:
- Authors can provide much more details about the technology including some successive figures of the technological flow. What SOI technique was used for 5nm, 7nm.
Reply: The conventional MOSFET is a surface channel device. In long channel conventional MOSFETs technology, the characteristics of transistor were at par with the essential speed as well as power requirements. In this era of electronics, power saving and low leakage are more crucial compared to increase in speed. The drive current rises with the new generation of transistor. However, there is also a tremendous enhancement in subthreshold leakage current, which results in increase in the power consumption [1]. Moreover, in small channel length FETs, the depletion regions underneath the source and drain cause degraded FET’s off-state leakage (IOFF), poor sub-threshold slope (SS), and threshold voltage (VT) reduction by drain-induced barrier lowering. To overcome those short channel effects, thin body thickness (Tbody) Si-on-Insulator (SOI) was invented. The SOI can use a substrate bias to improve the gate electrostatic control on channel carriers. When transistor sizes grew smaller in conventional planar MOSFETs with a 1.2 nm SiO2 gate oxide, the market required a significant innovation to retain performance while limiting short channel effects and power in advanced technologies as DC leakage in SiO2 is intolerably high. It was necessary to create a gate dielectric that could be substituted with SiO2, one that was thick enough to block direct electron tunneling through it but permeable enough to allow the electric field of the gate to enter the channel. Therefore, the solution was to use high dielectric-constant (high-κ) dielectric, a dielectric material that has a higher dielectric permittivity than SiO2. Although the use of high-κ dielectrics with metal gates increased the lifetime of planar MOSFET by another decades, it became necessary to introduce new devices beyond the 28 nm technology node to address the problems with traditional MOSFET [2]. Tremendous efforts have been made to reduce oxide thickness (tox) and increase εox to further decrease gate length while retaining sufficient gate controllability. Yet the Si Tbody of SOI requires continuously thinning down to improve the short channel effects, which cause a technology challenge. One simple solution is to form the three-dimensional FinFET that has even thinner Si Fin down to 6 nm Tbody thickness. Besides, both sidewalls and top surface of Fin are covered by gate-oxide and metal-gate, which have better gate electrostatic control of channel carriers than SOI. Therefore, the FinFET has been applied to 22 nm to 3 nm technology nodes, rather than using SOI.
(a) (b) (c)
Figure 1. The evolution of device structure from (a) planar MOSFET on bulk Si wafer, (b) planar MOSFET ultra-thin body SOI, and (c) 3D FinFET.
- Bohr, M.T.; Chau, R.S.; Ghani, T.; Mistry, K. The high-k solution. IEEE Spectr. 2007, 44(10), 29-35.
- Wong, H.S.P. Beyond the conventional transistor. Solid-State Electron. 2005, 49(5), 755-762.
- How was measured and correlated Qe charge and mobility ueff?
Reply: The µeff of FET are calculated according to the conventional metal-oxide-semiconductor (MOS) FET model [22-24]:
(1)
where LG and WG are the length and width of the conducting channel respectively, Cox is the gate-oxide capacitance.
The definition of Qe is
The µeff has been plotted as a function of Qe.
- Liu, Y.; Duan, X.; Shin, H.J.; Park, S.; Huang, Y.; Duan, X. Promises and prospects of two-dimensional transistors. Nature, 2021 591(7848), 43-53.
- Siao, M.D.; Shen, W.C.; Chen, R.S.; Chang, Z.W.; Shih, M.C.; Chiu, Y.P.; Cheng, C.M. Two-dimensional electronic transport and surface electron accumulation in MoS2. Nat. commun. 2018, 9(1), 1442.
- Takagi, S.I.; Toriumi, A.; Iwase, M; Tango, H. On the universality of inversion layer mobility in Si MOSFET's: Part I-effects of substrate impurity concentration. IEEE Trans. on Electron Dev. 1994, 41, 2357-2362.
- It is missing also a SEM/TEM image of the nano-transistor section to prove the thicknesses.
Reply: As suggested by the reviewer, the TEM image 5 nm-SnON/SiO2/HfO2 stack has been included in the revised manuscript to prove the thickness.
Figure 2 displays the cross-sectional transmission electron microscopy (TEM) image of the 5 nm-SnON/SiO2/HfO2 stack on Si substrate. A nanocrystalline uniform SnON layer of 5 nm ultra-thin thickness was observed. To enlarge the ION, gate insulator with high-κ [18] HfO2 was employed to reduce the operating voltage. Between the channel and gate dielectric, SiO2 with 3 nm thickness was deposited to limit the remote phonon scattering occurring from high-κ gate dielectric [19].
Figure 2. TEM image of the 5 nm-SnON/SiO2/HfO2 stack.
- Yu, X.; Zhu, C.; Yu, M.; Li, M.F.; Chin, A.; Tung, C.H.; Gui, D; Kwong, D.L. Advanced MOSFETs using HfTaON/SiO2/gate dielectric and TaN metal gate with excellent performances for low standby power application. In IEEE International Electron Devices Meeting, IEDM Tech. Dig. 2005, pp. 27-30.
- Fischetti, M.V.; Neumayer, D.A; Cartier, E.A. Effective electron mobility in Si inversion layers in metal–oxide–semiconductor systems with a high-κ insulator: The role of remote phonon scattering. J. Appl. Phys. 2001, 90(9), 4587-4608.
- In Introduction, authors claim the FinFET evolution till 1-3nm. I request authors to inset also o brif discussion about another exponent - the NOI transistor citing Ref: (i) Vacuum nano-triode in Nothing-On-Insulator configuration working in Terahertz domain, IEEE Journal of the Electron Devices Society, 6(1), pp. 1115-1123, 2018.(ii) Manufacturing of a Nothing On Insulator Nano-Structure with two Cr/Au Nanowires Separated by 18 nm Air Gap, Nanotechnology,31(27), pp.1-9, 2020,
https://dx.doi.org/10.1088/1361-6528/ab7c45
Reply: Thanks for the nice suggestion. We add the following sentences in the revised manuscript:
The continuous downscaling decreases the transistor’s the source and drain distance and causes lowered drain-voltage (VD) and power consumption of VDID/2, where ID is the drain-current. The ultimate VD downscaling is limited by the voltage drop in sub-threshold region, which has an idea SS of 60 mV/dec. Although the SS can be improved by using the charges in ferroelectric gate dielectric [3], the relatively large thickness and crystallized high-κ gate dielectric are the major concerns to integrate into highly scaled FinFET and nanosheet FET.
On the other hand, high VD is required to deliver enough output power for wireless communication [4]. The highly scaled FinFET and nanosheet FET cannot sustain the high VD that will cause the device to breakdown. Fortunately, the Vacuum Nano-Triode device in Nothing-On-Insulator (NOI) configuration may overcome this challenge by operating at a relatively high VD [5, 6]. This transistor showed excellent performance up to 4 THz that is crucial for six generation (6G) wireless communication. For logic application, further research and development to lower the VD and VG less than 1 V is required for NOI transistor.
- Cheng, C.H.; Chin, A. Low-voltage steep turn-on p-MOSFET using ferroelectric high-κ gate dielectric. IEEE Electron Device Lett. 2014, 35(2), 274-276.
- Chang, T.; Kao, H.L.; Chen, Y.J.; Liu, S.L.; McAlister, S.P.; Chin, A. A CMOS-compatible, high RF power, asymmetric-LDD MOSFET with excellent linearity. in IEEE International Electron Devices Meeting (IEDM) Tech. Dig., pp. 457-460, San Francisco, Dec. 2008.
- Ravariu, C. Vacuum nano-triode in nothing-on-insulator configuration working in terahertz domain. IEEE J. Electron Devices Soc. 2018, 6, 1115-1123.
- Ravariu, C.; Pârvulescu, C.; Manea, E.; Dinescu, A.; Gavrila, R.; Purica, M; Arora, V. Manufacture of a nothing on insulator nano-structure with two Cr/Au nanowires separated by 18 nm air gap. Nanotechnology, 2020, 31(27), 275203.
After these brief revision I recommend the paper publishing.
Thanks for the constructive comments.

Reviewer 2 Report
This paper reports high mobility in 5 nm thick SnON FETs obtained experimentally and supported by certain theoretical considerations. I think that the explanations concerning Coulomb scattering on charged traps are adequate, but I am not convinced by the explanations given for the high charge densities case. The results are interesting and potentially useful, but there are a series of objections that need to be addressed before the paper can be reconsidered for publication.
MAJOR:
(1) Previous paper by the same authors reports the same devices, and also the high 300 cm2/Vs mobility. The authors must make clear what are the specific novel results in this manuscript in comparison to the published paper: Pooja, P., Che, C. C., Zeng, S.-H., Lee, Y. C., Yen, T.-J., Chin, A., Outstanding High Field-Effect Mobility of 299 cm2 V−1 s−1 by Nitrogen-Doped SnO2 Nanosheet Thin-Film Transistor. Adv. Mater. Technol. 2023, 8, 2201521. https://doi.org/10.1002/admt.202201521
(2) The mobility is the central topic, so it is surprising that transport modeling/simulation is not done, nor relevant references studied, compared and cited. Takagi's classic paper is cited, but it would be meaningful and appropriate to cite relevant work about mobility modeling/experiments in ultra-thin-body semiconductors such as InGaAs and some 2D materials such as MoS2, WS2, all mentioned in the introduction, e.g.: DOI: 10.1016/j.sse.2015.07.007, DOI: 10.1109/TED.2012.2189217
(3) Equation (1) is most likely incorrect for ultra-thin layers, as it is derived for bulk semiconductors. Additionally, the only way to find what mechanism is responsible for slower mobility degradation is by physics-based modeling, which is not done in this paper. Please check and cite relevant literature/books on semiclassical transport specifically in ultra-thin layers, e.g. Esseni, D., Palestri, P., & Selmi, L. (2011). Nanoscale MOS Transistors: Semi-Classical Transport and Applications. Cambridge: Cambridge University Press. doi:10.1017/CBO9780511973857
(4) Please explain the derivation and adequacy of equation (2). Is it derived for 2D systems such as your devices? The literature usually reports expressions for 3D bulk semiconductors, so the authors need to be careful.
(5) Explanation including orbitals on p. 6, lines 172-174, is unclear. Currently I do not see how the radius of s-orbitals is related to higher mobility in SnON FETs. This issue needs clarifications and additional discussion.
MINOR:
(6) When discussing effective mass on p. 3, it is unclear how were they extracted, and which carrier type they belong to. Additionally, low effective mass is beneficial for long channel devices, but are detrimental for short-channel FETs due to enhanced tunneling.
(7) For SnO2 FETs in Fig. 2a, why is Vd biased only up to 1V, in contrast to 2.5V for other devices?
(8) In FETs the drain current is usually normalized by channel width and expressed in mA/um. This allows the comparison between different technologies, device architectures, materials, etc.
(9) Large on-off ratio, mentioned on p. 4, is important for digital logic applications; the devices reported in this paper have very long channels so it is questionable to assess their usage in future ICs where much shorter gate length are necessary.
(10) Why is 1/mobility plotted? What new information is provided by the inverse of the mobility?
Minor changes needed.
Author Response
Dear Reviewer 2,
We have responded to your important comments, point by point, and have included them in the revised manuscript (shown in green color). Your comments and our replies are in the following:
MAJOR:
(1) Previous paper by the same authors reports the same devices, and also the high 300 cm2/Vs mobility. The authors must make clear what are the specific novel results in this manuscript in comparison to the published paper: Pooja, P., Che, C. C., Zeng, S.-H., Lee, Y. C., Yen, T.-J., Chin, A., Outstanding High Field-Effect Mobility of 299 cm2V−1s−1 by Nitrogen-Doped SnO2 Nanosheet Thin-Film Transistor. Adv. Mater. Technol. 2023, 8, 2201521. https://doi.org/10.1002/admt.202201521
Reply: The major findings beyond our previous published paper [14] are the much lower µeff decay rate at high Eeff than SiO2/Si, high-κ/InGaAs, high-κ/2D MoS2 nFETs etc. This is the new discovery that was never reported in any FET device. The preserved high µeff at high Qe is critical to deliver for FET a high transistor’s output current and drive the IC speed quickly.
The high µeff is due to the >10× higher κ value of SnO2 than other semiconductor materials of Si, GaAs, InP, GaN and SiC, which can lower the channel effective field (Eeff) by >10× even at high Qe. Besides, the small 0.29 mo effective mass (me*), large overlapped s-orbitals and low phonon scattering may also play important roles to increase the mobility, although the µeff depends on both extrinsic and intrinsic scattering mechanism and will be discussed in following sessions.
- Pooja, P.; Che, C.C.; Zeng, S.H.; Lee, Y.C.; Yen, T.J.; Chin, A. Outstanding high field‐effect mobility of 299 cm2V−1s−1 by nitrogen‐doped SnO2 nanosheet thin‐film transistor, Adv. Mater. Technol. 2023, 2201521.
(2) The mobility is the central topic, so it is surprising that transport modeling/simulation is not done, nor relevant references studied, compared and cited. Takagi's classic paper is cited, but it would be meaningful and appropriate to cite relevant work about mobility modeling/experiments in ultra-thin-body semiconductors such as InGaAs and some 2D materials such as MoS2, WS2, all mentioned in the introduction, e.g.: DOI: 10.1016/j.sse.2015.07.007, DOI: 10.1109/TED.2012.2189217
Reply: We have added the references above in the revised manuscript [43, 44].
Because the remarkably high µeff SnON nFET is the new data, there is no modeling on the experimental data reported so far. In the scientific society of semiconductor device, the experiment is done before the mobility modeling. The modeling work following experiments can be evident from past high mobility InGaAs nFET development. The superb µeff in a 5 nm ultra-thin Tbody will attract modeling experts in the following up works.
- Poljak, M.; Jovanovic, V.; Grgec, D.; Suligoj, T. Assessment of electron mobility in ultrathin-body InGaAs-on-insulator MOSFETs using physics-based modeling. IEEE trans. on electron devices, 2012, 59(6), 1636-1643.
- Gonzalez-Medina, J.M.; Ruiz, F.G.; Marin, E.G.; Godoy, A.; Gámiz, F. Simulation study of the electron mobility in few-layer MoS2 metal–insulator-semiconductor field-effect transistors. Solid-State Electron. 2015,114, 30-34.
(3) Equation (1) is most likely incorrect for ultra-thin layers, as it is derived for bulk semiconductors. Additionally, the only way to find what mechanism is responsible for slower mobility degradation is by physics-based modeling, which is not done in this paper. Please check and cite relevant literature/books on semiclassical transport specifically in ultra-thin layers, e.g. Esseni, D., Palestri, P., & Selmi, L. (2011). Nanoscale MOS Transistors: Semi-Classical Transport and Applications. Cambridge: Cambridge University Press. doi:10.1017/CBO9780511973857
Reply: We rewrite this Equation.
(3)
This equation is basically the Gauss’s law. The Gauss law is one of the Maxwell’s equations [37], which cannot be changed in ultra-thin Tbody device. This is exactly the reason why this equation has been widely used for 2D material FETs [38].
The carrier transport in ultra-thin body or 2D materials is determined by both intrinsic mobility of phonon scattering and Coulomb scattering from the charge impurities and defects, and extrinsic effects of remote phonon scattering from high-κ dielectrics, and surface roughness scattering from the oxide/semiconductor interfaces. For InGaAs nFET at a relatively thick Tbody larger than 20 nm [40], the μeff is dominated by intrinsic phonon scattering. Therefore, the μeff of InGaAs nFET is higher than that of Si due to the smaller me*. However, at thin InGaAs Tbody less than 20 nm, the extrinsic scattering of interface defects limits the μeff [40]. In this report, the µeff at thin Tbody of 7 and 5 nm are still higher than that of Si and InGaAs nFET. The reason can only be ascribed to the superb intrinsic property of >10× smaller Eeff to lower the interface scattering, smaller me*, and high phonon limited mobility.
It is well known that the device modeling is developed after MOSFET fabrication in IC industry, such as the widely used Berkeley Short-channel IGFET Model (BSIM). In this model, there are many fitting parameters to be measured experimentally, in addition to physically based equations. As the devices become smaller in each technology node by Moore's law, new versions of device models are developed to accurately reflect the transistor's behavior. Therefore, the BSIM model changes continuously for the past three more decades. Such device modeling requires years long experience from both academic and IC fabs’ team works, which is beyond our group’s capability. Similar device modeling followed by this record high µeff nFET may be developed later by theoreticians that are typical for the past InGaAs FET [40-43] and 2D materials FETs [44, 45].
- 37. Maxwell's equations. Available online: https://en.wikipedia.org/wiki/Maxwell%27s_equations (accessed on 8, June, 2023)
- Liu, Y.; Duan, X.; Huang, Y; Duan, X. Two-dimensional transistors beyond graphene and TMDCs, Chem. Soc. Rev. 2018, 47, 6388-6409.
- Alamo, J. del; Vardi, A.; Zhao, X. InGaAs FinFETs for future CMOS. Invited Paper, Compound Semiconductor Magazine, August/September 2016, pp. 22-26.
- Alamo, J. A. del. CMOS extension via III-V compound semiconductors. Short Course on Emerging Nanotechnology and Nanoelectronics at IEEE International Electron Devices Meeting, Washington, DC, December, 2007, pp. 10-12.
- Alamo, J. A. del.; Kim, D. H. InGaAs CMOS: a "Beyond-the-Roadmap" Logic Technology? Invited paper presented at 2007 Device Research Conference, University of Notre Dame, South Bend, IN, June 18-20, 2007, pp. 201-202.
- Poljak, M.; Jovanovic, V.; Grgec, D.; Suligoj, T. Assessment of electron mobility in ultrathin-body InGaAs-on-insulator MOSFETs using physics-based modeling. IEEE trans. on electron devices, 2012, 59(6), 1636-1643.
- Gonzalez-Medina, J.M.; Ruiz, F.G.; Marin, E.G.; Godoy, A.; Gámiz, F. Simulation study of the electron mobility in few-layer MoS2 metal–insulator-semiconductor field-effect transistors. Solid-State Electron. 2015,114, pp.30-34.
- Esseni, D.; Palestri, P.; Selmi, L. Nanoscale MOS transistors: semi-classical transport and applications. Cambridge: Cam-bridge University Press. 2011.
(4) Please explain the derivation and adequacy of equation (2). Is it derived for 2D systems such as your devices? The literature usually reports expressions for 3D bulk semiconductors, so the authors need to be careful.
Reply:
In the revised manuscript, the previous eq. (2) is now eq. (4).
In ultrathin body 2D materials, carrier transport is determined by phonon scattering from the dielectrics and coulomb scattering from charged defects such as vacancies [51]. Ma and Jena et al. predicted that high-κ dielectrics provide effective screening of the charge impurities leading to high Coulomb-limited mobility [52]. Moreover, owing to the low formation energy of the chalcogen vacancy, a large amount of sulfur vacancies is commonly observed in synthesized 2D MoS2, which can induce short-range scattering and degrade carrier mobility [53]. Besides, excellent matching of equation (4) with measurements are also reported for SnO2 nFET with a 5 nm channel thickness [46].
- Shih, C.W.; Chin, A. New material transistor with record-high field-effect mobility among wide-band-gap semiconductors. ACS Appl. Mater. Interfaces 2016, 8, 19187–19191.
- Das, S.; Sebastian, A.; Pop, E.; McClellan, C.J.; Franklin, A.D.; Grasser, T.; Knobloch, T.; Illarionov, Y.; Penumatcha, A.V.; Appenzeller, J.; Chen, Z. Transistors based on two-dimensional materials for future integrated circuits. Nat. Electron. 2021, 4(11), 786-99.
- Ma, N.; Jena, D. Charge scattering and mobility in atomically thin semiconductors. Phys. Rev. 2014, 4, 011043.
- Qiu, H.; Xu, T.; Wang, Z.; Ren, W.; Nan, H.; Ni, Z.; Chen, Q.; Yuan, S.; Miao, F.; Song, F.; Long, G.Hopping transport through defect-induced localized states in molybdenum disulphide. Nat. commun. 2013, 4(1), 2642.
(5) Explanation including orbitals on p. 6, lines 172-174, is unclear. Currently I do not see how the radius of s-orbitals is related to higher mobility in SnON FETs. This issue needs clarifications and additional discussion.
Reply: The theoretical background of high mobility in metal-oxide semiconductor is due to the overlapped s-orbitals [54]. The larger s-orbitals are, and the stronger overlapping of electron clouds lead to high mobility. We have earlier reported that in SnON, the localized states just above the valence band maximum (VBM) have predominant N 2p character and the lower conduction states near conduction band minimum (CBM) were mostly derived from Sn 5s orbitals which results in high electron mobility in SnON [14].
- Pooja, P.; Che, C.C.; Zeng, S.H.; Lee, Y.C.; Yen, T.J.; Chin, A. Outstanding high field‐effect mobility of 299 cm2V−1s−1 by nitrogen‐doped SnO2 nanosheet thin‐film transistor, Adv. Mater. Technol. 2023, 2201521.
- Wei S. C.; Chin A.; Fu L. C.; Fang S. W. Remarkably high mobility ultra-thin-film metal-oxide transistor with strongly overlapped orbitals. Sci. Rep. 2016 6(1), 19023.
MINOR:
(6) When discussing effective mass on p. 3, it is unclear how were they extracted, and which carrier type they belong to. Additionally, low effective mass is beneficial for long channel devices, but are detrimental for short-channel FETs due to enhanced tunneling.
Reply: The effective mass was cited from our previous paper of E-k diagrams [14].
Figure r. Energy band structures of (a) SnO2 and (b) SnON [14]
The searching for high µeff material nFET leads to extensive research on high mobility InGaAs nMOSFET [41]. The reason why this material failed to implement into manufacture is due to relatively inferior oxide/semiconductor interface and caused µeff degradation in thin Tbody, rather than the enhanced tunneling. For Tbody less than 20 nm, the gm and gm/ Tbody of Si FinFET are still better than those of InGaAs FinFET [40].
- Pooja, P.; Che, C.C.; Zeng, S.H.; Lee, Y.C.; Yen, T.J.; Chin, A. Outstanding high field‐effect mobility of 299 cm2V−1s−1 by nitrogen‐doped SnO2 nanosheet thin‐film transistor, Adv. Mater. Technol. 2023, 2201521.
- Alamo, J. del; Vardi, A.; Zhao, X. InGaAs FinFETs for future CMOS. Invited Paper, Compound Semiconductor Magazine, August/September 2016, pp. 22-26.
- Alamo, J. A. del. CMOS extension via III-V compound semiconductors. Short Course on Emerging Nanotechnology and Nanoelectronics at IEEE International Electron Devices Meeting, Washington, DC, December, 2007, pp. 10-12.
(7) For SnO2 FETs in Fig. 2a, why is Vd biased only up to 1V, in contrast to 2.5V for other devices?
Reply: To make the reading consistent, we have changed all the ID-VD plots to 1 V. The ID-VG is more important to calculate µeff-Qe.
|
|
|
(a) |
|
|
|
|
|
(b) |
(c) |
|
Figure 4. ID-VD output characteristics for (a) TaN/HfO2/5-nm-SnO2 nFET (b) TaN/HfO2/5-nm-SnON nFET and (c) TaN/HfO2/7-nm-SnON nFET.
(8) In FETs the drain current is usually normalized by channel width and expressed in mA/um. This allows the comparison between different technologies, device architectures, materials, etc.
Reply: For accurate µeff extraction, a fat FET (long channel FET) [21] made in IC fabs must be used to lower the difference between physical and electrical gate length, where the source and drain depletion regions can decrease the electrical gate length. This is the reason why mA is used for Y-axis, rather than mA/μm.
- Sun, S.C.; Plummer, J.D. Electron mobility in inversion and accumulation layers on thermally oxidized silicon surfaces. IEEE J. of solid-state circuits, 1980. 15(4), 562-573.
(9) Large on-off ratio, mentioned on p. 4, is important for digital logic applications; the devices reported in this paper have very long channels so it is questionable to assess their usage in future ICs where much shorter gate length are necessary.
Reply: In the InGaAs FET [40]-[42] and 2D material FET [56] evolution, long gate length device was first made to investigate the intrinsic property, such as µeff, Ion/Ioff and SS. The downscaling InGaAs nFET took a decade long study, until the µeff degraded fast with decreasing ultra-thin Tbody. After the record high µeff is reported, researchers and engineers in IC fabs will follow up to study the small gate length devices and the potential to be implanted in the gate all around (GAA) nanosheet FET.
- Alamo, J. del; Vardi, A.; Zhao, X. InGaAs FinFETs for future CMOS. Invited Paper, Compound Semiconductor Magazine, August/September 2016, pp. 22-26.
- Alamo, J. A. del. CMOS extension via III-V compound semiconductors. Short Course on Emerging Nanotechnology and Nanoelectronics at IEEE International Electron Devices Meeting, Washington, DC, December, 2007, pp. 10-12.
- Alamo, J. A. del.; Kim, D. H. InGaAs CMOS: a "Beyond-the-Roadmap" Logic Technology? Invited paper presented at 2007 Device Research Conference, University of Notre Dame, South Bend, IN, June 18-20, 2007, pp. 201-202.
- Qian, Q.; Lei, J.; Wei, J.; Zhang, Z.; Tang, G.; Zhong, K.; Zheng, Z.; Chen, K.J., 2D materials as semiconducting gate for field-effect transistors with inherent over-voltage protection and boosted ON-current. npj 2D Materials and Applications, 2019, 3(1), 24.
(10) Why is 1/mobility plotted? What new information is provided by the inverse of the mobility?
Reply: The 1/µeff has a linear relationship with Qe.
1/µeff = kQe, (2)
where k is the proportional constant.
The inversely linear relationship between µeff and Qe is exactly the same as the µeff dependence on ionized impurity concentrations [24]. This confirms that the charged Vo in SnO2 is the major reason to cause Coulomb scattering.
- Takagi, S.I.; Toriumi, A.; Iwase, M; Tango, H. On the universality of inversion layer mobility in Si MOSFET's: Part I-effects of substrate impurity concentration. IEEE Trans. on Electron Dev. 1994, 41, 2357-2362.
Thanks for the constructive comments.

Round 2
Reviewer 2 Report
My comments have been addressed appropriately. Copy-editing and proofing team will have to format parts of the text, equations, figures, etc. However, on the technical content side, the revised version is in much better shape and I can recommend it for publication.
English seems fine, minor editing needed.